# Robust Feature Matching with Spatial Smoothness Constraints

**Xu Huang [1,2,*], Xue Wan [3,4] and Daifeng Peng [5,6]**

1    School of Geospatial Engineering and Science, Sun Yat-Sen University, Zhuhai 519082, China
2    Wuhan Engineering Science and Technology Institute, Wuhan 430019, China
3    Key Laboratory of Space Utilization, Chinese Academy of Sciences, Beijing 100094, China; wanxue@csu.ac.cn
4    Technology and Engineering Center for Space Utilization, Chinese Academy of Sciences,
     Beijing 100094, China
5    School of Remote Sensing and Geomatics Engineering, Nanjing University of Information Science and
     Technology, Nanjing 210044, China; daifeng@nuist.edu.cn
6    The Department of Information Engineering and Computer Science, University of Trento, 38123 Trento, Italy
*    Correspondence: huangx358@mail.sysu.edu.cn; Tel.: +86-1597-220-7433

**Abstract:** Feature matching is to detect and match corresponding feature points in stereo pairs, which is one of the key techniques in accurate camera orientations. However, several factors limit the feature matching accuracy, e.g., image textures, viewing angles of stereo cameras, and resolutions of stereo pairs. To improve the feature matching accuracy against these limiting factors, this paper imposes spatial smoothness constraints over the whole feature point sets with the underlying assumption that feature points should have similar matching results with their surrounding high-confidence points and proposes a robust feature matching method with the spatial smoothness constraints (RMSS). The core algorithm constructs a graph structure from the feature point sets and then formulates the feature matching problem as the optimization of a global energy function with first-order, spatial smoothness constraints based on the graph. For computational purposes, the global optimization of the energy function is then broken into sub-optimizations of each feature point, and an approximate solution of the energy function is iteratively derived as the matching results of the whole feature point sets. Experiments on close-range datasets with some above limiting factors show that the proposed method was capable of greatly improving the matching robustness and matching accuracy of some feature descriptors (e.g., scale-invariant feature transform (SIFT) and Speeded Up Robust Features (SURF)). After the optimization of the proposed method, the inlier number of SIFT and SURF was increased by average 131.9% and 113.5%, the inlier percentages between the inlier number and the total matches number of SIFT and SURF were increased by average 259.0% and 307.2%, and the absolute matching accuracy of SIFT and SURF was improved by average 80.6% and 70.2%.

**Keywords:** feature matching; SIFT; SURF; Delaunay triangulation; global energy function; spatial smoothness constraint

## 1. Introduction

Feature matching is to detect and find corresponding feature points in stereo pairs, which is an important prerequisite in camera orientation with the basic knowledge that the optical rays from the corresponding feature points should intersect at the same object point [1,2]. Though being studied for decades, feature matching is still one of the most popular topics in photogrammetry and computer vision communities. Its great contributions in camera orientation have fueled many photogrammetric and computer vision applications, e.g., 3D reconstruction [3–5], motion capture [6,7], virtual reality [8,9], image registration [10,11], change detection [12,13] and large-scale mapping [14,15].

Most feature matching algorithms firstly detect feature points with obvious intensity variations, then describe the intensity features of these points in the local matching windows, and finally, find the corresponding points with the most similar feature descriptors [16,17]. However, some factors may limit the feature matching accuracy, e.g., large viewing angles of stereo cameras, and resolution differences of stereo pairs. These limiting factors will cause serious geometric distortions in the matching windows of the feature points, thus may bring uncertainties in their descriptors. To address this issue, most work either rectifies stereo pairs/matching windows to reduce the geometric distortions or designs a certain geometric-distortion-invariant feature descriptor in stereo matching.

In the former case, either stereo images or matching windows are firstly rectified with the given initial orientation parameters so that some geometric distortions (e.g., scale and rotation) can be corrected, and feature points are then matched in the rectified pairs or matching windows. Therefore, such rectification-based methods can be categorized into 1) the image rectification-based methods and 2) the matching window rectification-based methods. The image rectification-based methods normally project both stereo images onto a common height plane with the given initial orientation parameters (e.g., Rational Polynomial Coefficient (RPC) parameters in the satellite dataset [18–20], position and attitude parameters in the aerial dataset [21,22]). Each pixel in the rectified images is geo-coded with the height plane so that some geometric distortions (e.g., scale and rotation) can be corrected except for the ones caused by the terrain elevation differences. On the other hand, the window rectification-based methods instead of geo-code matching windows with the given orientation parameters so that some geometric distortions within the windows can be corrected. They either project multi-view matching windows onto a common height plane [23] or adjust the window shapes according to the epipolar lines [24,25]. In general, these rectification-based methods are simple and efficient in reducing some geometric distortions for robust feature matching. However, such methods cannot be applied when no orientation parameters are available, which is a common case in close-range photogrammetry and computer vision. Besides, such methods cannot correct the geometric distortions that are caused by the terrain elevation differences.

To achieve robust matching results without any available orientation parameters, some work designs robust feature descriptors against some geometric distortions, among which the scale-invariant feature transform (SIFT) [26] descriptor is perhaps the most famous. SIFT firstly detects feature points in the difference of Gaussian (DOG) pyramids so that the scale differences between stereo images can be greatly reduced, and then computes orientations of each feature point in the matching windows, which are used to correct rotations between stereo images. Therefore, SIFT is a useful scale- and rotation-invariant feature descriptors, while its high-dimensional descriptors will bring high time complexity. For more efficient matching, Bay et al. [27] designed a much lower-dimensional feature descriptor through Haar wavelet without scarifying its robustness against scale- and rotation-distortions (also termed Speeded Up Robust Features, SURF). Leutenegger et al. [28] proposed a novel scale-space FAST-based detector in combination with a bit-string descriptor (also termed BRISK), which can further speed up the feature matching when compared with SIFT and SURF. To apply SIFT in more complex geometric distortions, Morel and Yu [29] rectified stereo pairs by considering all possible camera views and found the best camera views with the most SIFT matches. Such a method has been proven to be affine invariant, thus also termed Affine-SIFT (ASIFT). Alcantarilla et al. [30] improved the feature detection part of SIFT by using nonlinear diffusion filtering, which could detect more robust feature points. In recent years, several convolutional neural network (CNN) based methods [31,32] have been developed to extract deep features from matching windows, which can achieve more robust and more accurate matching results than those traditional low-level feature (e.g., gradients and binary strings) based methods. However, all the above feature matching methods only compute the local optima of the matching of each feature point. They did not consider feature matching more globally. Therefore, the uncertainties in the matching results of each feature point cannot be further reduced. In addition to the geometric distortions, image textures also influence the final matching results. In weak or repeat

texture regions, all the above local matching methods could not achieve robust matching results, due to the low-intensity features and the high matching uncertainties.

To improve the local feature matching methods, some work introduced spatial constraints to improve the feature matching results. Li et al. [33] firstly selected reliable matches or seed points by a novel region descriptor, then assumed local affine transformations among these seed points, and finally reduced the corresponding search range of other feature points under these local affine-transformation constraints. Chen et al. [34] selected matches with higher distinctiveness as seed matches, defined second-order geometric transformation based on the seed matches to reduce the corresponding search range of other features, and then found correspondences for all the features within the reduced searching range. Ma et al. [35] solved for correspondence by interpolating a vector field between the two feature point sets and imposed nonparametric geometrical constraints in the vector field to obtain good matches. Winter et al. [36] imposed reliable descriptor co-occurrences to improve the matching performances. Such methods normally compute more accurate and more robust matching results than those local matching methods, while their results are partly scene dependent. Some large geometric distortions may reduce the reliability of seed point selections as well as the matching guidance of the spatial constraints.

To further improve the robustness and the accuracy of feature matching, especially in large geometric-distortion regions and weak/repeat texture regions, this paper optimizes the matching results of each feature point by leveraging the matching results of its surrounding feature points with the underlying constraints that feature points should have similar matching results to their surrounding high-confidence points and proposes a novel robust matching method with the spatial smoothness constraints (RMSS). The core algorithm considers the feature matching more globally, and formulates the feature matching problem as the optimization of a graph-based energy function with the spatial smoothness constraint. For efficient computation purposes, the global optimal solution of the energy function is then broken into sub-optimizations, where the matching result of each feature point is iteratively optimized by its surrounding feature points. Therefore, the uncertainties (caused by some geometric distortions or weak/repeat textures) in the matching results of feature points can be greatly reduced, and the matching robustness and the matching accuracy can be significantly improved, especially in some large-distortion scenarios and weak/repeat texture scenarios. In addition to the novel mathematical principles of the proposed method, the main contributions of the proposed method include: (1) some spatial constraints based matching (SCM) methods correct unreliable matches by seed points, while the proposed method instead correct them by the optimization of an energy function, thus avoiding the mismatches caused by wrong seed points; (2) some SCM methods impose specific spatial constraints among sparse seed points and may provide inaccurate matching clues (especially in disparity jumps), while the proposed method introduces spatial smoothness constraints between denser adjacent feature points, thus being fit for more complex scenes. The experiments on close-range datasets show that the proposed method can greatly increase the feature matching accuracy.

## 2. Methodology

### 2.1. Workflow

Given a pair of stereo images $\{I_L, I_R\}$ with $I_L, I_R$ being the left and the right images in the pair, and the corresponding feature point sets $\{P_L, P_R\}$ with $P_L, P_R$ being the sets of feature points in the left and the right images through a certain feature matching algorithm (e.g., SIFT or SURF in this paper), traditional feature matching methods find the correspondence in the right with the most similar descriptor for each feature in the left. However, due to some geometric distortions or weak/repeat textures, the correspondences with the most similar descriptors may not be true. Such uncertainties may greatly reduce the final matching accuracy as well as the robustness.

To reduce such uncertainties, this paper simultaneously considers several potential correspondences for each feature in the basic image (e.g., the left), and imposes spatial smoothness constraints across the

whole feature point set to iteratively optimize the matching result of each feature. The core algorithm formulates the feature matching as the optimization of a global energy function based on a graph, which takes each feature point as nodes and defines the spatial relationships (i.e., edges) of these nodes through Delaunay meshing [37]. The solut7ion of the energy function is the final matching result of the proposed method. In general, the workflow of the proposed method follows an iterative manner (Figure 1): (1) detect feature points in stereo images, compute their intensity features using a certain feature descriptor (e.g., SIFT and SURF in this paper), and find several potential correspondences for each feature in the basic through KD-tree [38] by ranking their distances in the feature space, as shown in Figure 1a; (2) formulate these distances in the feature space as the matching cost terms of the energy function, and each feature in the basic has a series of matching cost with respect to its potential correspondences, as shown in Figure 1b, where the maximum cost among the potential correspondences is only 1.23 times larger than the minimum cost; (2) define the spatial relationships of these feature points through Delaunay meshing (as shown in Figure 1c), impose smoothness constraints between adjacent feature points, and formulates these constraints as the smoothness terms of the energy function, (3) break the optimization of the global energy function into sub-optimizations of each feature point, and iteratively compute the local solutions of these sub-optimizations so that the matching uncertainties in the matching cost can be reduced, as shown in Figure 1d, where the second minimum cost is 4.45 times larger than the minimum cost; (4) take the other image as the basic, repeat step (1) to (3), and detect and eliminate mismatches through the left-right-consistency (LRC) strategy. The final remaining matching points are shown in Figure 1e.

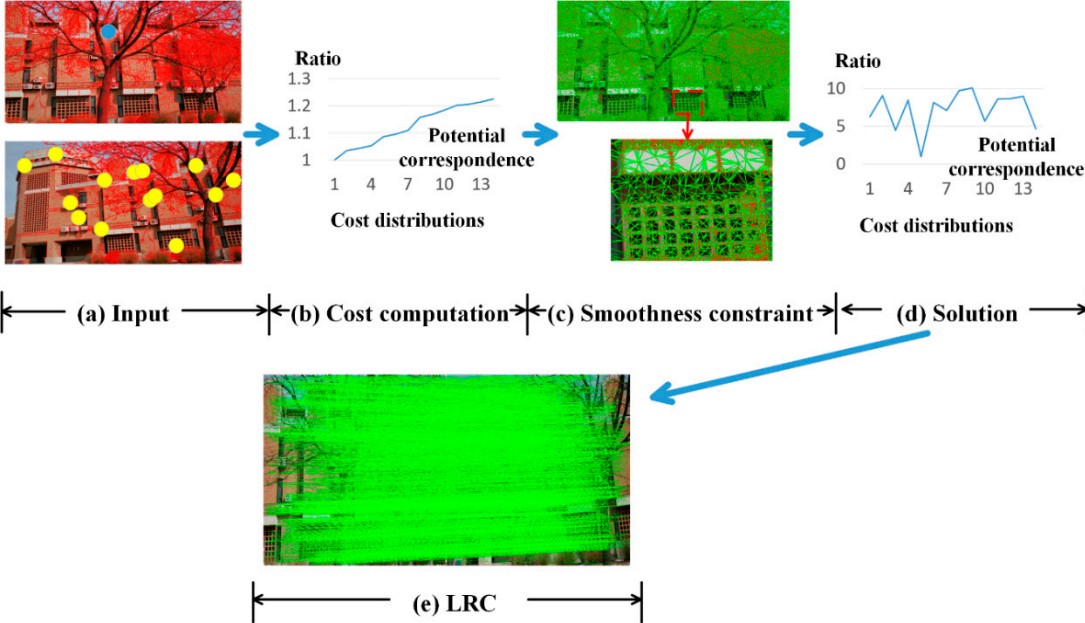

**Figure 1.** The workflow of the proposed method. In (**a**), red dots show the feature points in stereo pairs, the blue and the yellow circles represent a feature point in the basic image and its potential correspondences in the other, respectively. In (**b**) and (**d**), the horizontal axis represents the index of the potential correspondences, and the vertical axis represents the ratio between the matching cost at a certain correspondence index and the minimum cost of the feature point; in (**c**), the red dots are feature points, and the green edges of the triangle meshes define the spatial relationships of these feature points; in (**e**), the green lines connect matches of the proposed method after the LRC strategy.

## 2.2. Feature Point Graph

To impose the spatial smoothness constraints in the feature matching, each feature point needs to first find its surrounding feature points. To achieve this goal, Delaunay meshing is used to connect all feature points in the triangularization form, as shown in Figure 2a,b. The feature points and the edges in the Delaunay mesh construct a graph $\mathbf{G} = (\mathbf{V}, \mathbf{E})$ with $\mathbf{V}$ being the set of nodes/feature points and $\mathbf{E}$ being the set of edges. The matching result optimization of each feature point will be influenced by its connected feature points. However, not all connected feature points can be used for matching optimization. For example, some feature points have high uncertainties in their matching cost, thus may propagate the uncertainties to other feature points through the edges. To address this issue, the edges starting from the high-uncertainty/low-confidence feature points should be limited, and the edges starting from the low-uncertainty/high-confidence feature points should be connected, thus constructing a bidirectional graph (Figure 2c), where each edge has two weights with respect to different directions. The weights of each edge will be described in Section 2.3. Each node in the graph has a series of potential corresponding points, as shown in Figure 2d. Therefore, the final point feature graph is a 3D discrete, irregular, bidirectional volume.

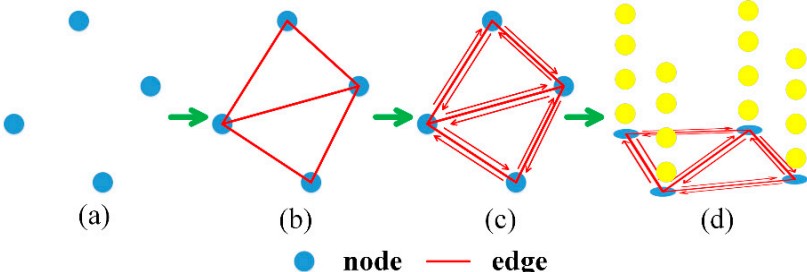

Figure 2. Feature point graph. Blue circles are feature points or nodes in the graph; red lines are edges in the graph; yellow circles are potential corresponding points. (**a**) is the set of feature points with blue circles being nodes in the graph; (**b**) is the Delaunay meshing results from these feature points with red lines being edges in the graph; (**c**) shows bidirectional edges between feature points; (**d**) shows the potential corresponding points (the yellow circles) of each feature point.

## 2.3. Graph-Based Energy Function

To find good corresponding points from the potential ones, this paper imposes the spatial smoothness constraints between any two connected feature points with the underlying assumptions that feature points should have similar matching results to their surrounding high-confidence points, and formulates the feature matching problem as the minimization of an energy function based on the feature point graph $\mathbf{G} = (\mathbf{V}, \mathbf{E})$. In general, the energy function consists of two terms: (1) cost term, which measures the feature similarity between feature points and their potential corresponding points, and (2) smoothness term, which penalizes the inconsistent matching results between any two connected feature points. The formulation of the energy function is as follows.

$$M(\mathbf{L}) = \sum_{p \in \mathbf{V}} C\big(p, l_p\big) + \sum_{e_{q,p} \in \mathbf{E}} P\big(e_{q,p}\big) \cdot \big\|d_p - d_q\big\|_2 / \big\|p - q\big\|_2$$
$$d_p = \mathbf{V}'\big(l_p\big) - p$$
$$d_q = \mathbf{V}'\big(l_q\big) - q$$

(1)

where, $M$ is the energy function; $\mathbf{L}$ is the set of matching results (i.e., the correspondence ID) of all feature points in the basic image; $C\big(p, l_p\big)$ is the matching cost of the feature point $p$ at its potential correspondence label $l_p$; $\mathbf{V}'$ is the set of feature point in the other image; $\mathbf{V}'\big(l_p\big)$ is the potential corresponding point of $p$ at label $l_p$; $\|\cdot\|_2$ measures the vector norm; $d_p$ measures the disparities of $p$ in

both row and column directions; $e_{q,p}$ is an edge from $q$ to $p$; $P(e_{q,p})$ is the weight of the edge $e_{q,p}$, which is also termed as the penalty coefficient. It means the contribution of $q$ in the matching of $p$. Since $\mathbf{G}$ is a bidirectional graph, $P(e_{q,p})$ and $P(e_{p,q})$ often have different values, which helps to reduce the influence of low-confidence points on the matching optimization. To improve the matching robustness in slanted regions, the disparity difference $\left\|d_p - d_q\right\|_2$ is divided by $\left\|p - q\right\|_2$ for the purpose of the normalization.

In this paper, the cost term $C(p, l_p)$ computes the feature similarity between the feature point $p$ and its potential corresponding point $\mathbf{V}'(l_p)$ by measuring their distance in the feature space. There are various feature descriptors, among which SIFT [26] and SURF [27] are adopted in the experimental comparisons in Section 4. However, the sizes of different feature descriptors are not the same, thus leading to different matching cost ranges. To increase the generality of the proposed method, this paper normalizes the matching cost by dividing it by the maximum matching cost, as follows.

$$C(p, l_p) = \left\| f(p) - f\left(\mathbf{V}'(l_p)\right) \right\|_2 / C_{max}$$
$$C_{max} = \max_{i \in N_{range}} \{ C(p, l_i) \}$$
(2)

where, $f$ is a certain feature descriptor (e.g., SIFT or SURF in this paper); $C_{max}$ is the maximum matching cost among the matching cost of $p$; $N_{range}$ is the number of potential correspondences. The final feature matching optimization results of the proposed method partly depend on the number of potential correspondences. Too small a number may exclude the true label, while too large a number significantly increases the time cost of the feature matching. In this paper, the appropriate correspondence number will be analyzed in Section 4.1.

The smoothness term guides the matching results of any two connected feature points consistent by penalizing the inconsistent cases. The matching result consistency is measured by the distances between their disparity vectors in both row and column directions. Shorter distances mean more consistent matching results as well as smaller penalties, and vice versa. However, some feature points with low matching confidence may give bad influences on the matching of its surrounding feature points. Therefore, the penalty coefficient $P$ of edges starting from these low-confidence feature points should be reduced. In addition, the penalty coefficient $P$ is also related to the spatial distance between the connected feature points. Considering matching in slanted regions, larger distance should correspond to smaller $P$, and vice versa. Therefore, the penalty coefficient $P(e_{q,p})$ is a function with two independent variables: (1) the matching confidence of $q$ and (2) the spatial distance between $p$ and $q$, as follows:

$$P(e_{q,p}) = P_0 \cdot \left(a + r_q\right)^b / \left\|p - q\right\|_2$$
$$r_q = 1 - C_1^{min} / C_2^{min}$$
(3)

where, $P_0$ is a predefined initial penalty coefficient; $r_q$ is the matching confidence of $q$, which is formulated as the ratio between the minimum matching cost $C_1^{min}$ and the second minimum matching cost $C_2^{min}$; $a$, $b$ are predefined parameters, which are used to increase the penalty coefficient of high-confidence points and decrease the ones of low-confidence points. $a$, $b$ adopt fixed values in all experiments with $a = 0.4$ and $b = 3$. $P_0$ defines the scale of the penalty coefficient. In this paper, the appropriate values of $P_0$ will be analyzed in Section 4.2.

## 2.4. Solution

The solution of the global energy function in Equation (1) is the final feature matching result. However, the optimization of Equation (1) is a typical NP-hard problem, which means that its global solution has high time complexity. For efficient computation purposes, a compromise solution of Equation (1) is computed by splitting the global optimization into a collection of sub-optimizations

and then iteratively computing the local minimum for the matching of each feature point. To achieve this goal, Equation (1) is first transformed into the summation of the local functions in Equation (4).

$$M(\mathbf{L}) = \sum_{p \in \mathbf{V}} U(l_p \mid p, \mathbf{E}_p)$$

$$U(l_p \mid p, \mathbf{E}_p) = C(p, l_p) + \sum_{e_{q,p} \in \mathbf{E}_p} P(e_{q,p}) \cdot \|d_p - d_q\|_2 / \|p - q\|_2 \tag{4}$$

where, $\mathbf{E}_p$ is the set of edges that only end at $p$; $U(l_p \mid p, \mathbf{E}_p)$ is a sub-optimization function which only considers the cost term of $p$ and the smoothness term of edges that ends at $p$.

For computational purposes, this paper only computes the local solution of the sub-optimization $U(l_p \mid p, \mathbf{E}_p)$ instead of the global optimal solution of $M(\mathbf{L})$, i.e., transform the energy function $\min M(\mathbf{L})$ into $\sum_{p \in \mathbf{V}} \min U(p, l_p, \mathbf{E}_p)$, which can greatly reduce the variable space. In the sub-optimization of $U(l_p \mid p, \mathbf{E}_p)$, the matching result of the connected points in $\mathbf{E}_p$ is assumed available so that the matching of $p$ is constrained by the connected, high-confidence points, as follows.

$$l_p = \operatorname{argmin} U(l_p \mid p, \mathbf{E}_p) = \operatorname{argmin}\left\{ C(p, l_p) + \sum_{e_{q,p} \in \mathbf{E}_p} P(e_{q,p}) \cdot \|d_p - d_q^0\|_2 / \|p - q\|_2 \right\} \tag{5}$$

where, $d_q^0$ is the initial matching result of the connected point $q$.

The matching result of $p$ partly depends on $d_q^0$. However, due to some geometric distortions and weak/repeat textures, the initial matching result $d_q^0$ may be unreliable. To further improve the matching accuracy, the sub-optimization in this paper proceeds in an iterative manner: (1) find the corresponding point, for each feature point in the basic image, with the minimum matching cost as the initial matching results; (2) compute the local optima of the matching for each feature point by considering the initial matching results of their connected points, as shown in Equation (5); (3) update the initial matching results and the matching cost $C$ with the local optima and the more reliable cost $U$, respectively, and then detect the number of inliers from the optimal solutions by using the fundamental matrix; (4) repeat step (2) to (3) until the inlier number in the current iteration is smaller than the one in the previous iteration; (5) output the matching results with the most inliers. The iterative sub-optimization manner is summarized as the pseudo-code in Algorithm 1.

To remove mismatches in the matching results, this paper adopts the left-right-consistency check (LRC) strategy. In general, this paper respectively takes the left and the right images as the basic finds the corresponding points for each feature in the basic images by using the proposed method, compares these two matching results, and detects and removes the inconsistent part whose matching inconsistency is larger than 2 pixels. The matching inconsistency is measured by computing the distance between the original feature point in the left and the matching result of its corresponding point in the right, as follows.

$$\delta = \|p - p'\|_2$$
$$q = \mathbf{V}'(l_p) \quad p' = \mathbf{V}(l_q) \tag{6}$$

where, $p$ is a feature point in the left; $q$ is the corresponding point of $p$ when the left is the basic image; $l_q$ is the correspondence label of $q$ when the right is the basic image; $p\prime$ is the corresponding point of $q$; $\delta$ is the distance between $p$ and $p\prime$.

---

**Algorithm 1:** The iterative solution of the sub-optimization in the proposed method

---

**Function:** Iterative_solution_of_sub_optimizations ($\mathbf{P}_B$, $\mathbf{P}_{Other}$, $\mathbf{N}$, $\mathbf{C}$, $\mathbf{L}$)
**Input**: the set of feature point in the basic image $\mathbf{P}_B$, the set of feature point in the other image $\mathbf{P}_{Other}$, the set of connected points for each feature point in the basic image $\mathbf{N}$; the set of matching cost of all feature point $\mathbf{C}$;
**Output**: the set of matching results of all feature points in the basic image $\mathbf{L}$

---

**Pseudo-code:**

1. //step1: find the initial correspondence with the minimum matching cost
2. **for** $i$ =1 to Num, **do** //Num is the feature point number in the basic
3. $\mathbf{L}_0(i) = \underset{l}{\operatorname{argmin}} \mathbf{C}(i, l)$ //$\mathbf{L}_0(i)$ is the initial matching result of the feature point $i$
4. **end for**
5. inlier_num_crt = Detect_inliers_using_Fundamental_matrix ($\mathbf{P}_B$, $\mathbf{P}_{Other}$, $\mathbf{L}_0$)
6. inlier_num_before = 0
7. inlier_num_max = inlier_num_crt
8. $\mathbf{L} = \mathbf{L}_0$
7. while (inlier_num_crt > inlier_num_before)
8. {
9. //step2: compute the local optima of the matching in Equation (5)
10. **for** $i$ =1 to Num, **do**
11. //$\mathbf{L}_{refine}(i)$ is the new matching result of $i$ constrained by its surrounding connected points $\mathbf{N}_i$
12. //$\mathbf{U}$(i) is the optimized matching cost after the local optimization
13. [$\mathbf{L}_{refine}(i)$, $\mathbf{U}$(i)] = Local_optima_of_matching ($\mathbf{C}$, $i$, $\mathbf{N}_i$, $\mathbf{L}_0(\mathbf{N}_i)$)
14. **end for**
15. //step3: update the matching results as well as the matching cost
16. $\mathbf{L}_0 = \mathbf{L}_{refine}$
17. $\mathbf{C} = \mathbf{U}$
18. inlier_num_before = inlier_num_crt
19. inlier_num_crt = Detect_inliers_using_Fundamental_matrix ($\mathbf{P}_B$, $\mathbf{P}_{Other}$, $\mathbf{L}_0$)
20. //update the output matching result
21. **if** (inlier_num_crt > inlier_num_max)
22. inlier_num_max = inlier_num_crt
23. $\mathbf{L} = \mathbf{L}_0$
24. **end if**
25. }

---

## 3. Study Areas and Data

The proposed method was tested on close-range datasets provided by the National Laboratory of Pattern Recognition, Institute of Automation, Chinese Academy of Science, including the Tsinghua gate dataset, Bioscience building dataset, Fayu temple dataset, and the Zhantan temple dataset [39]. All datasets were captured by Canon EOS 5D. To evaluate the matching performance of the proposed method, especially in large geometric-distortion and weak/repeat texture regions, this paper respectively selected three pairs from each dataset with large scale differences (e.g., Figure 3a1,b2,c1), large perspective distortions (e.g., Figure 3a2,b3,c3,d1,d2), occlusions by pedestrians, trees or other man-made objects (Figure 3a3,c2,d2) and weak/repeat textures (Figure 3b,c). The limiting factors in most pairs are the combinations of multiple above distortion and texture factors. To evaluate the matching accuracy of the proposed method, this paper manually selected nine matching points as checking points for each image pair. The distributions of checking points on each image pair are shown as red circles in Figure 3.

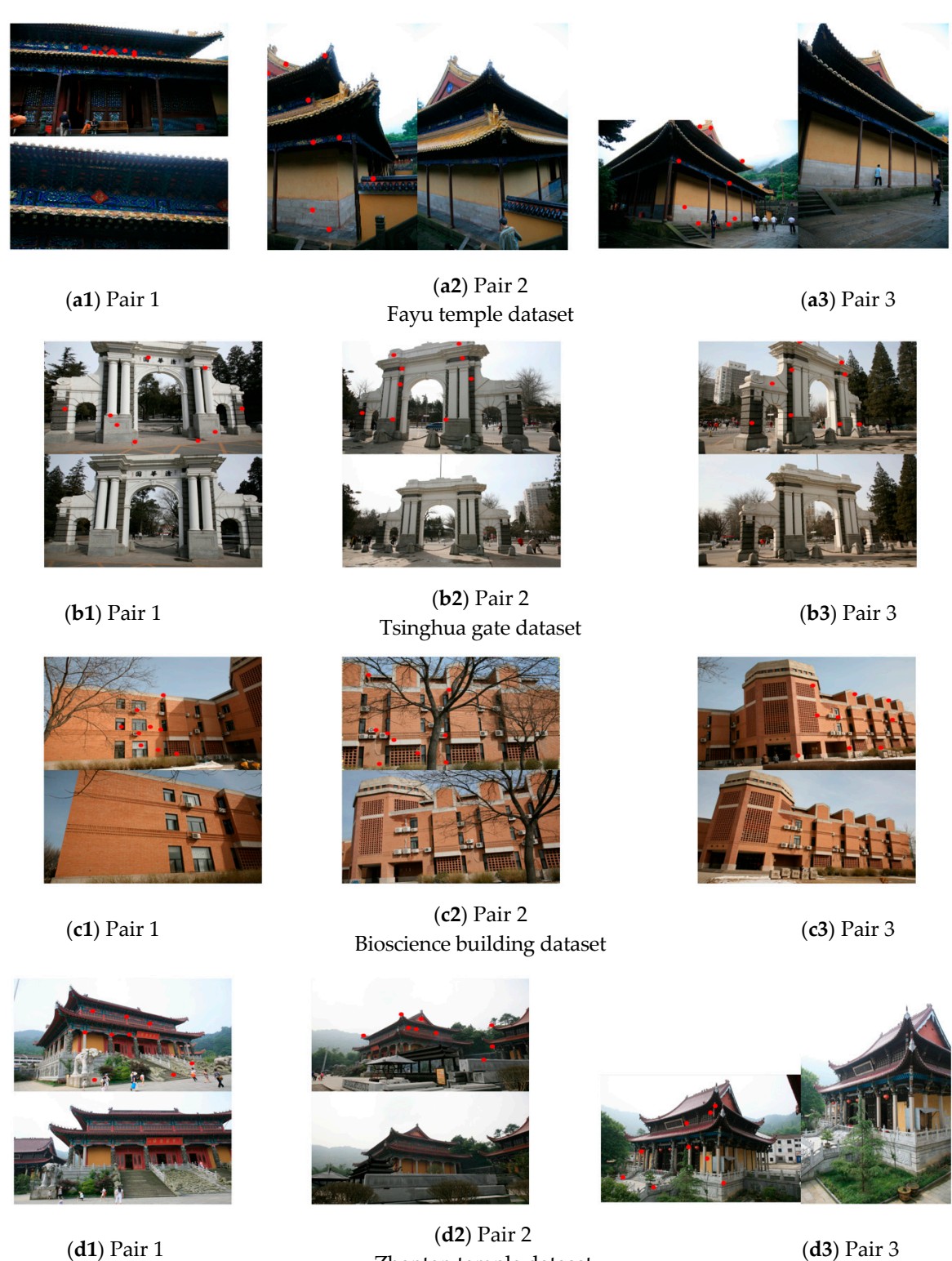

(**a1**) Pair 1

(**a2**) Pair 2
Fayu temple dataset

(**a3**) Pair 3

(**b1**) Pair 1

(**b2**) Pair 2
Tsinghua gate dataset

(**b3**) Pair 3

(**c1**) Pair 1

(**c2**) Pair 2
Bioscience building dataset

(**c3**) Pair 3

(**d1**) Pair 1

(**d2**) Pair 2
Zhantan temple dataset

(**d3**) Pair 3

**Figure 3.** The close-range datasets for experimental analysis and comparisons.

## 4. Results

The proposed method was used to optimize the matching results of SIFT and SURF, termed SIFT + RMSS and SURF + RMSS, respectively. For fair comparisons, all matching methods (SIFT, SURF, SIFT + RMSS, and SURF + RMSS) utilized the LRC strategy to detect and eliminate the mismatches, and the remaining matching points were used in the experimental analysis and comparisons. The matching

accuracy of the proposed methods as well as other feature matching methods was evaluated in three aspects: (1) the number of inliers $n_{inliers}$, which measures the robustness of the matching methods, especially in large-geometric-distortion regions and weak/repeat texture regions, (2) the percentage of inliers $per_{inlier}$, which measures the uncertainties in the matching results of each method, and (3) the matching accuracy against the checking points $acc_{chk}$ (also termed absolute accuracy). Matching points of each method were first utilized to compute a fundamental matrix through a RANSAC strategy [40]. The number of inliers $n_{inliers}$ is counted by computing the epipolar lines of matching points through the fundamental matrix and adopting the matching points with the distances to the epipolar lines smaller than 2 pixels as inliers. The percentage of inliers $per_{inlier}$ is then formulated as the ratio between the number of inliers and the number of all matching points. On the other hand, the absolute matching accuracy using checking points $acc_{chk}$ was formulated as the average distance to the epipolar lines of all checking points.

In general, this paper firstly analyzed some parameters of the proposed method on the Fayu temple dataset, i.e., the number of the potential correspondences in Section 4.1 and the initial penalty coefficient in Section 4.2, by studying the matching accuracy (including the inlier number $n_{inliers}$, the inlier percentage $per_{inlier}$ and the absolute matching accuracy $acc_{chk}$) with various parameters, and found the most appropriate parameters with the highest matching accuracy. Then, the proposed method was used to optimize the matching results of SIFT and SURF on the Tsinghua gate dataset, Bioscience building dataset, and Zhantan temple dataset, and the optimized matching results were compared with the original matching results as well as a robust matching method via vector field consensus (VFC) [35] in the aspects of $n_{inliers}$, $per_{inlier}$ and $acc_{chk}$ in Section 4.3. VFC solved for correspondence by interpolating a vector field between the two feature point sets and imposed nonparametric geometrical constraints in the vector field to obtain good matches. All experiments were conducted on the same compute configuration with a single CPU @ 2.60GHz.

*4.1. Analysis about the Influence of the Number of the Potential Correspondences on the Final Matching Results*

The matching results of the proposed method partly depend on the number of potential correspondences (i.e., $N_{range}$ in Equation (2)). Too small a number may exclude the true correspondence, while too large a number will increase the time cost. To select appropriate number of the potential correspondences, this paper firstly utilized the proposed method to optimize the matching results of SIFT on the Fayu temple dataset and then analyzed the matching accuracy of the proposed method in the metrics of the inlier number $n_{inliers}$, the inlier ratio $per_{inlier}$ and the absolute matching accuracy $acc_{chk}$ with various $N_{range}$ (from 2 to 20). As the ranges of the metrics in different pairs may be significantly different (e.g., some pairs have hundreds of inliers, while some only have dozens of inliers), this paper normalized the matching accuracy of the proposed method by dividing it by the matching accuracy of SIFT (termed $n_{inliers}^0$, $per_{inlier}^0$ and $acc_{chk}^0$, respectively) so that the normalized matching accuracy on different pairs were at the same level, as shown in Figure 4. $n_{inliers}/n_{inliers}^0$ in Figure 4a and $per_{inlier}/per_{inlier}^0$ in Figure 4b are the ratio between the number of inliers of the proposed method and SIFT, and the ratio between the percentages of inliers of the proposed method and SIFT, respectively, thus the larger ratio $n_{inliers}/n_{inliers}^0$ (> 1) and $per_{inlier}/per_{inlier}^0$ (> 1) mean more robust matching results. $acc_{chk}/acc_{chk}^0$ in Figure 4c is the ratio between the absolute matching accuracy of the proposed method and SIFT, thus smaller ratio $acc_{chk}/acc_{chk}^0$ (< 1) means higher matching accuracy.

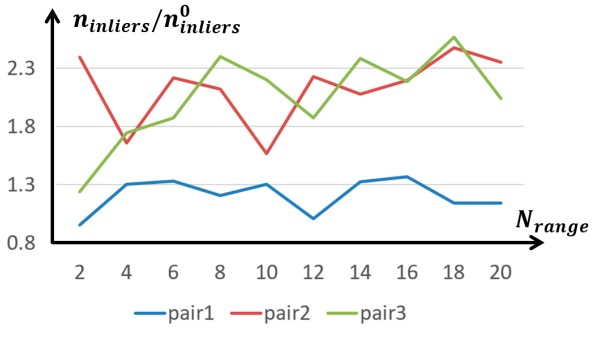

(**a**) Number of inliers

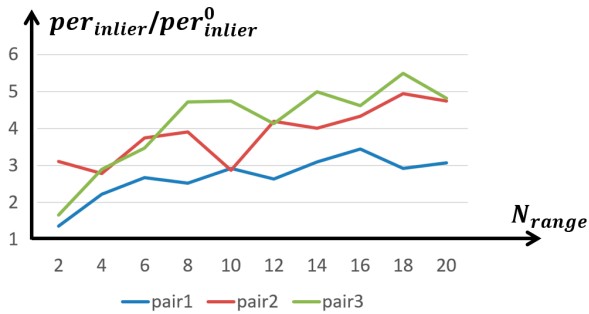

(**b**) Percentage of inlier

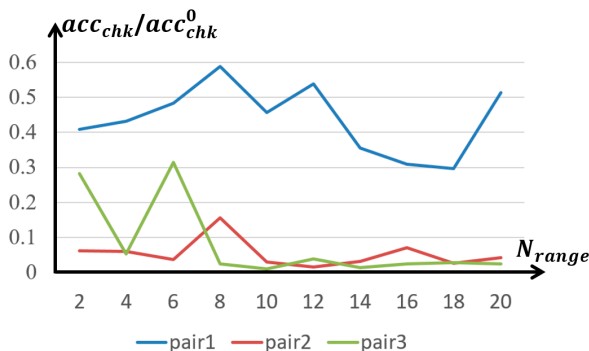

(**c**) Absolute matching accuracy

**Figure 4.** The trend of matching accuracy with various $N_{\text{range}}$ on the Fayu temple dataset. (**a**) shows the inlier number trend with increasing $N_{\text{range}}$; (**b**) shows inlier percentage trends with increasing $N_{\text{range}}$; (**c**) shows the absolute matching accuracy trends with increasing $N_{\text{range}}$.

In Figure 4a, almost all number ratios $n_{inliers}/n_{inliers}^0$ on all pairs are larger than one (except for pair 1 with $N_{range}$ as two), which means that the inlier number of SIFT has been increased after the optimization of our proposed method. In Figure 4b, all percentage ratios $per_{inliers}/per_{inliers}^0$ are larger than one, which means that the inlier percentages of the proposed methods were also higher than SIFT. Therefore, the proposed method can provide more robust feature matching results for the later camera orientations. The percentage ratio $per_{inliers}/per_{inliers}^0$ has an increasing trend with the increasing of $N_{\text{range}}$, since larger $N_{\text{range}}$ means a lower probability of selecting the same matching results for the mismatches. More robust matching results (higher inlier number and higher inlier percentage) can often be used to compute higher-accuracy camera orientation results, thus all absolute accuracy ratio $acc_{chk}/acc_{chk}^0$ in Figure 4c are smaller than 1. Figure 4 shows that the proposed method could achieve higher matching accuracy, though $N_{\text{range}}$ varied greatly. To achieve the highest matching accuracy, the most appropriate $N_{\text{range}}$ is needed to be selected in the feature matching. However, different pairs

corresponded to different appropriate $N_{\text{range}}$ in different matching accuracy metrics, e.g., the most appropriate $N_{range}$ of pair 1 are 16, 16, and 18, respectively, in the metrics of $n_{inliers}$, $per_{inlier}$ and $acc_{chk}$, while the most appropriate ones of pair 2 are 18, 18, and 12, respectively, in the metrics of $n_{inliers}$, $per_{inlier}$ and $acc_{chk}$. To give a comprehensive evaluation of $N_{range}$, this paper, therefore, ranked the matching accuracy of each pair, as shown in Equation (7), and then summarized these ranks to find the most appropriate $N_{\text{range}}$ with respect to the highest ranking, as shown in Figure 5.

$$Rank_{i,j}\left(N_{range}\right) = \begin{cases} acc_{i,j}^{\max} / acc_{i,j}\left(N_{range}\right) & j = n_{inliers} \text{ or } per_{inlier} \\ acc_{i,j}\left(N_{range}\right) / acc_{i,j}^{\min} & j = acc_{chk} \end{cases} \tag{7}$$

where, $i$ represents a certain stereo pair; $j$ represents a certain matching accuracy metric; $acc_{i,j}\left(N_{range}\right)$ means the matching accuracy of pair $i$ in the metric of $j$ at a certain $N_{\text{range}}$; $Rank_{i,j}\left(N_{range}\right)$ means the rank of pair $i$ in the metric of $j$ at a certain $N_{\text{range}}$; $acc_{i,j}^{\max}$ means the maximum value of matching accuracy among various $N_{\text{range}}$, while $acc_{i,j}^{\min}$ means the minimum value of matching accuracy among various $N_{\text{range}}$. $Rank_{i,j}$ closer to one means higher matching accuracy. The summation of all ranking results is shown as follows.

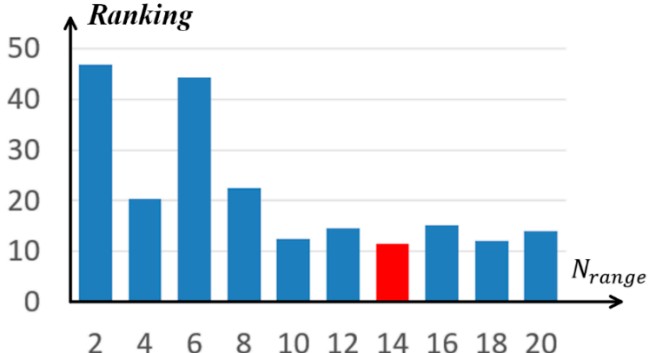

**Figure 5.** Ranking with various $N_{range}$ on the Fayu temple dataset.

In Figure 5, a shorter-length bin means higher ranking. The ranking results did not change much when $N_{\text{range}} \geq 10$, which shows that such a number of potential correspondences is sufficient to include the true correspondence. The red bin with $N_{\text{range}}$ as 14 has the highest ranking. Therefore, this paper adopted $N_{\text{range}} = 14$ in all the following experiments in Sections 4.2 and 4.3.

*4.2. Analysis about the Influence of the Initial Penalty Coefficient on the Final Matching Results*

In addition to the number of the potential correspondences $N_{\text{range}}$, this paper also evaluated the influences of the initial penalty coefficient (i.e., $P_0$ in Equation (3)) on the feature matching accuracy. Too small $P_0$ will bring weak smoothness constraints so that the matching of each feature point cannot be optimized by its surrounding points. On the other hand, too large $P_0$ will decrease the contribution of the matching cost in the feature matching process so that some mismatches will be generated in some slanted regions or height jumps. To select appropriate $P_0$, this paper still utilized the proposed method to optimize the matching results of SIFT on the Fayu temple dataset with various $P_0$ (from 0.02 to 0.2), and then analyzed the matching accuracy of the proposed method in the metrics of the inlier number $n_{inliers}$, the inlier ratio $per_{inlier}$ and the absolute matching accuracy $acc_{chk}$. For better experimental comparisons, this paper also normalized the matching accuracy of the proposed method by dividing it by the matching accuracy of SIFT (termed $n_{inliers}^{0}$, $per_{inlier}^{0}$ and $acc_{chk}^{0}$, respectively). The experimental comparisons on various $P_0$ are shown in Figure 6.

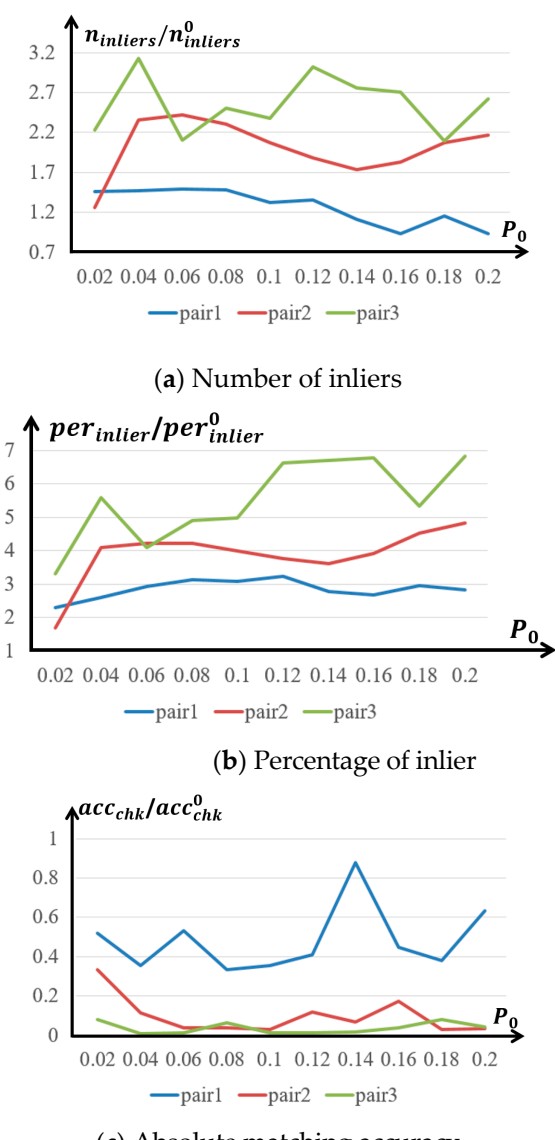

**(a)** Number of inliers

**(b)** Percentage of inlier

**(c)** Absolute matching accuracy

**Figure 6.** The trend of matching accuracy with various $P_0$ on the Fayu temple dataset. (**a**) shows the inlier number trend with increasing $P_0$; (**b**) shows inlier percentage trends with increasing $P_0$; (**c**) shows the absolute matching accuracy trends with increasing $P_0$.

In Figure 6a, most inlier number ratios $n_{inliers}/n^0_{inliers}$ are larger than one (except for pair 1 with $P_0$ as 0.16 and 0.2, respectively), which shows that the matching results of the proposed method is robust against the variation of $P_0$ when $P_0 < 0.16$. However, too large $P_0$ (e.g., $P_0 \geq 0.16$) may decrease the inlier number, since such large $P_0$ may bring over-smoothness constraints in the feature matching. In Figure 6b, all percentage ratios $per_{inliers}/per^0_{inliers}$ are larger than one, which means that the matching results of the proposed method contained more inliers as well as fewer outliers than SIFT. All absolute accuracy ratio $acc_{chk}/acc^0_{chk}$ in Figure 6c are smaller than 1, which means that the proposed method is capable of greatly improving the matching accuracy of SIFT. The highest matching accuracy improvement on the three pairs were 66.44% at $P_0 = 0.08$, 96.99% at $P_0 = 0.18$ and 98.89% at $P_0 = 0.04$, respectively. Therefore, the most appropriate $P_0$ for different pairs were different. To select the most appropriate $P_0$ in a general scenario, this paper still ranked the matching accuracy of each pair in Equation (7) and then summarized these ranks to find the most appropriate $P_0$ with respect to the highest ranking, as shown in Figure 7.

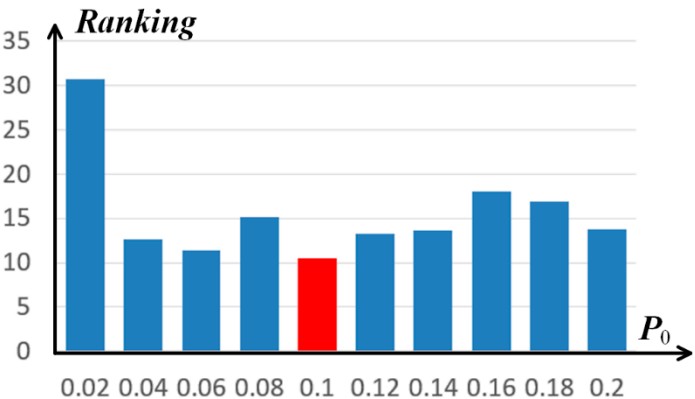

**Figure 7.** Ranking with various $P_0$ on the Fayu temple dataset.

In Figure 7, the ranking results did not change much when $0.04 \leq P_0 \leq 0.14$, which shows that such a range of the initial penalty coefficient is appropriate for the proposed method on the matching optimization of SIFT. The red bin with $P_0$ as 0.1 has the highest ranking. Therefore, this paper adopted $P_0 = 0.1$ in all the following experiments in 3.3.

*4.3. Matching Accuracy Comparisons on More Datasets*

To give more comprehensive evaluations of the proposed method, this paper applied SIFT [26] and SURF [27] on more datasets (i.e., Tsinghua gate dataset, Bioscience building dataset, and Zhantan temple dataset), utilized the proposed method to optimize the matching results of SIFT and SURF, respectively, and compared the proposed method with a state-of-the-art matching method VFC [35]. VFC improved the matching results of SIFT or SURF by interpolating a vector field between the two feature point sets and imposed nonparametric geometrical constraints in the vector field. The final matching results of VFC and the proposed method are termed as SIFT + RMSS, SURF + RMSS, SIFT + VFC, and SURF + VFC. All stereo pairs in the testing datasets were 4368×2912 pixels. Matching parameters of the proposed method were fixed by using the experimental conclusions in Sections 4.1 and 4.2, i.e., $N_{range} = 14$ and $P_0 = 0.1$. The matching accuracy of SIFT, SURF, SIFT + VFC, SURF + VFC, SIFT + RMSS, and SURF + RMSS were also evaluated in the three metrics, i.e., the inlier number $n_{inliers}$, the inlier ratio $per_{inlier}$ and the absolute matching accuracy $acc_{chk}$. The running time of these methods was also tested. The experimental comparisons about the matching accuracy and the running time are shown in Figure 8. The horizontal axes mean the ID of stereo pairs in the Tsinghua gate dataset, Bioscience building dataset, and Zhantan temple dataset in Figure 3, and the vertical axes mean the matching accuracy in Figure 8a–c, and the running time in Figure 8d. However, the ranges of the absolute accuracy of different matching methods on different pairs were significantly different. For better comparisons, this paper truncated absolute accuracy values that were larger than 100 pixels, and then divided all absolute accuracy values of a certain pair by ten as long as any one of the absolute accuracy values was larger than ten. Since the testing datasets were of poor matching quality (e.g., large geometric distortions, weak/repeat textures and occlusions) for SIFT and SURF, most pairs needed the absolute accuracy division operator except for the pair c-2 (marked by * in Figure 8c).

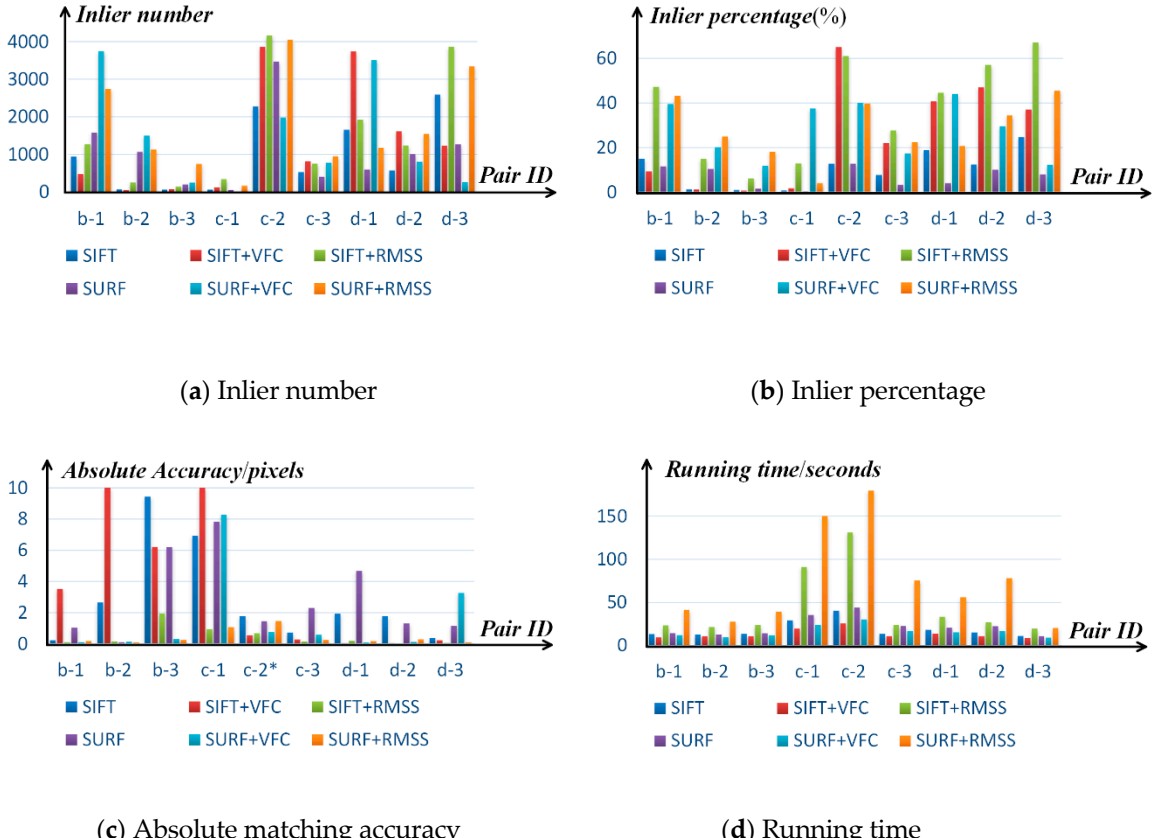

(**a**) Inlier number

(**b**) Inlier percentage

(**c**) Absolute matching accuracy

(**d**) Running time

**Figure 8.** Experimental comparisons among SIFT, SURF, VFC, and the proposed method

Figure 8a shows the inlier number of SIFT, SURF, SIFT + RMSS, and SURF + RMSS on all testing datasets. VFC generally increased the inlier number of SIFT and SURF by average 44.7% and 59.3%, respectively, while its matching results were not robust. In some cases (e.g., SIFT + VFC in b-1 and SURF + VFC in d-3), the inlier number after VFC was even less than the original ones. On the other hand, the inlier number of SIFT and SURF has been increased by the proposed method in all cases, which shows that the proposed method can robustly reduce the matching uncertainties in SIFT and SURF by imposing the smoothness constraints. After the optimization of the proposed method, the SIFT/SURF matching points can be increased by at most 437.5%/272.9%, at least 15.7%/5.5%, and average 131.9%/113.5%. In Figure 8b, the average inlier percentages of SIFT and SURF were only as low as 10.5% and 6.9%, respectively, due to the large geometric distortions, weak/repeat textures, and occlusions in the testing datasets. VFC can significantly improve the inlier percentages of SIFT and SURF to average 25.0% and 28.0%, respectively. However, VFC still met the unstable problem in some cases (e.g., SIFT + VFC in b-1), where the inlier percentages of VFC were lower than the original ones. The proposed method could improve the inlier percentage in all cases. After the optimization of the proposed method, the average inlier percentages of SIFT + RMSS and SURF + RMSS can be increased to 37.7% and 28.1%, respectively. It shows that the proposed method can also reduce the mismatches as well as increase the inliers in the matching results as the proposed method considers more potential correspondences in the matching and imposes the smoothness constraints to reduce the matching uncertainties. Though the general performance of VFC was not as good as the proposed method, it could achieve better results in some cases (e.g., c-2). In Figure 8c, the proposed method can significantly improve the absolute matching accuracy of SIFT/SURF by at most 97.9%/96.1%, at least 55.2%/−0.65%, and average 80.6%/70.2%. The proposed method is capable of improving the absolute matching accuracy in most cases except for the SURF matching of pair c-2. It is because the matching results of SURF on pair c-2 were already good with 3471 inliers, in which case the proposed method

did little to such a good matching result. The unstable inlier number and the inlier percentage of VFC resulted in unstable absolute matching accuracy. In some cases (e.g., SURF + VFC in b-3), it can significantly improve the absolute matching accuracy by 94.9%, while in some cases (e.g., SIFT + VFC in c-1), it might decrease the absolute matching accuracy by 44.1%. In general, the proposed method can greatly improve the matching results of SIFT and SURF on poor-quality stereo pairs. Since the proposed method optimized the matching results of SIFT and SURF in an iterative manner, the running time of the proposed method was certainly longer than the pure SIFT or the pure SURF. The proposed method added on average 25.3 s more time to the SIFT matching and 52.3 s more time to the SURF matching on the testing datasets. VFC was much more efficient than the proposed method with 61.3% and 73.0% less time cost for SIFT and SURF, respectively. It should be noted that VFC was even faster than pure SIFT and pure SURF. It is because VFC has already been capable of removing mismatches, therefore it could avoid the repeat computation of the LRC strategy. Considering only one single CPU is used in the proposed method, it has great potential to speed up when a parallel framework is applied.

To give more visual comparisons, this paper selected pairs with the least inliers from each testing dataset (i.e., (b-3), (c-1), and (d-2)), and illustrated the inliers of SIFT, SURF, SIFT + RMSS, and SURF + RMSS, respectively, as shown in Figure 9. Since Fayu temple dataset (Figure 3(a1)–(a3)) is a training dataset, which is used to train matching parameters of the proposed method, we did not illustrate the matching results on such dataset for fair comparisons.

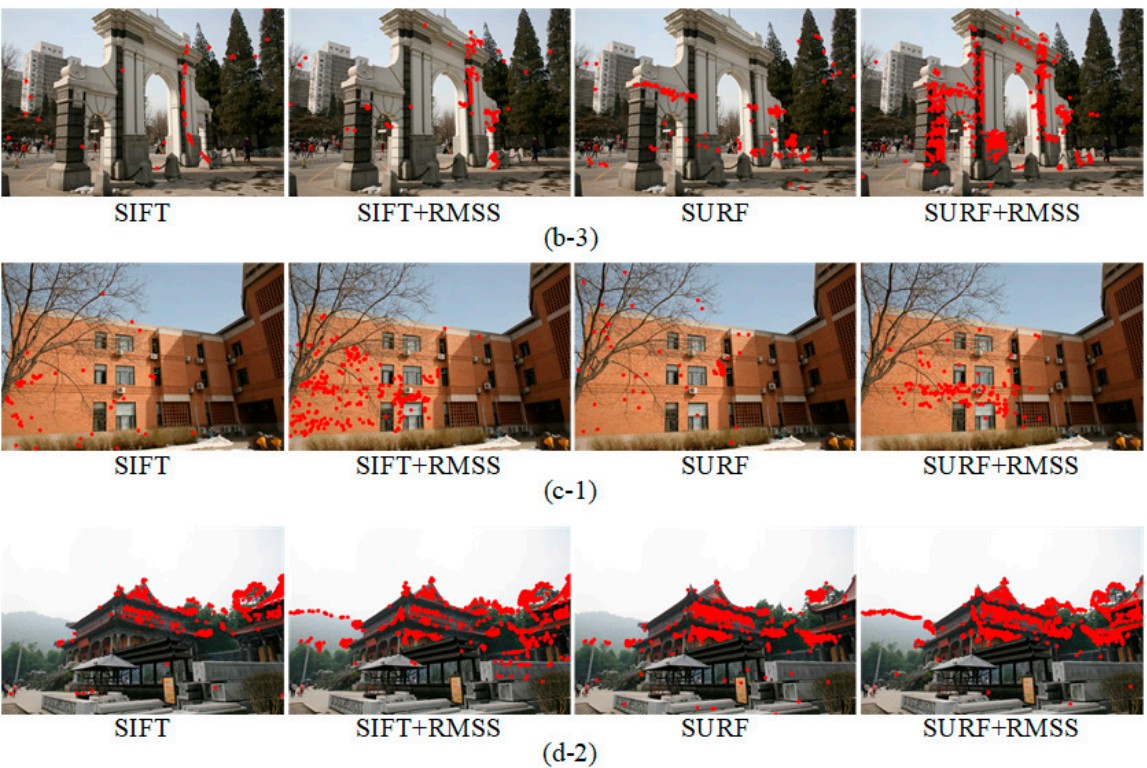

**Figure 9.** Inliers distributions before/after the optimization of the proposed method. (**b-3**) is the stereo pair with the least inliers in Tsinghua gate dataset; (**c-1**) is the stereo pair with the least inliers in Bioscience building dataset; (**d-2**) is the stereo pair with the least inliers in Zhantan temple dataset.

In Figure 9, the weak textures of the white wall in b-3, and the repeat textures of the black wall in (b-3) and the yellow wall in c-1 brought high matching uncertainties in SIFT and SURF. Thus, both methods only matched dozens of points. The proposed method-imposed smoothness constraints to reduce the matching uncertainties, thus being able to achieve more matching points in weak texture and repeat texture regions. In Figure 9d-2, the matching points of SIFT and SURF only concentrated on

the roof of the temple. The uneven distributions of matching points may lead to low camera orientation accuracy. The proposed method not only matched more points on the temple roof but also matched more even points (e.g., the points on the mountain). In general, the proposed method is capable of achieving more robust matching results than pure SIFT and pure SURF.

## 5. Discussion

The results of this study indicate that the proposed method can greatly improve the matching results of SIFT and SURF in some matching difficult regions (e.g., large geometric-distortion regions and repeat textured regions). The experimental comparisons were tested on the datasets of Tsinghua gate, Bioscience building, and Zhantan temple, which had serious matching limiting factors. Both SIFT and SURF performed badly on the testing datasets with the average inlier number being 973.7/1072.2, the average inlier ratio being 10.5%/6.9% and the average absolute matching accuracy being 27.0/27.5 pixels. As the proposed method reduces the matching uncertainties of feature points by imposing spatial smoothness constraints over the whole feature point sets, it can greatly improve the matching results of SIFT and SURF. After the optimization of the proposed method, the corresponding accuracy was improved to 1549.9/1759.8, 37.7%/28.1%, and 4.09/2.89 pixels, respectively. In addition to the matching accuracy improvement, the proposed method also obviously increased the matches of SIFT and SURF, as shown in Figure 9. For example, Figure 9b-3,c-1 have repeat textures on the walls of the Tsinghua gate and Bioscience building, where traditional SIFT and SUFR matched few points, due to the high matching uncertainties. The proposed method introduced spatial smoothness constraints to reduce these uncertainties, thus achieving more matches.

The study has gone some way towards enhancing our understanding of feature matching with spatial constraints. Before this, some spatial constraints-based matching (SCM) methods utilized seed points to constrain the matching process, while the wrong seed points will give wrong matching clues to the matching of other features, this may decrease the final matching accuracy. The proposed method instead constrains the matching process through the optimization of the energy function, thus being more fit for the matching in difficult regions. Therefore, the proposed method has a great potential to be applied in some close-range and oblique photogrammetric applications, e.g., street-view mapping, robot navigation, relic reconstruction, and city modeling.

The proposed method has some limitations to be considered. First, the parameter selections in Section 4.1 and 4.2 were determined through SIFT matching results and were also proved to be valid for SURF matching in Section 4.3. However, since the descriptors of different matching methods may be significantly different. The best parameters of SIFT and SURF may not be fit for other feature descriptors. Therefore, it is recommended to retrain new matching parameters if other feature descriptors were used. Second, the proposed method has a higher time cost than the pure SIFT and SURF, and the time differences have an increasing trend with the increasing of feature points. For example, the time cost of the proposed is 1.75 times higher than SIFT when the number of SIFT features is 56,165, while the multiple reaches 3.25 when the number of SIFT features is 167,159. The increasing multiple of time cost is caused by the Delaunay triangularization and the optimization of the energy function in the proposed method. Therefore, the proposed method was not fit for the scenarios with large amounts of feature points and high time efficiency demands. The above limiting factors of the proposed method need to be further investigated in the future. We plan to enhance the normalization of different feature descriptors so that the trained matching parameters could have more general applicability. We also plan to improve the efficiency of the spatial relationship determination of adjacent feature points by introducing tree-structure indexing.

## 6. Conclusions

This paper aims at addressing the matching uncertainties issues in large geometric-distortion regions, weak-texture regions, and repeat-texture regions, and proposes a robust feature matching method with spatial smoothness constraints to reduce such uncertainties with the underlying

assumptions that each feature point should have a similar matching result to its surrounding high-confidence points. The core algorithm formulates the feature matching as a minimization of a global energy function based on a feature point graph and introduces a comprise solution by breaking the optimization into a collection of sub-optimizations of each feature point. The proposed method is capable of greatly optimizing the matching results of SIFT and SURF on some poor-quality stereo pairs. Experiments on several poor-quality, close-range datasets show that the proposed method can significantly increase the inliers number of SIFT and SURF by average 131.9% and 113.5%, improve the inlier ratios of SIFT and SURF by average 259.0% and 307.2%, and improve the absolute matching accuracy of SIFT and SURF by average 80.6% and 70.2%, respectively. However, the proposed method will add more running time to the feature matching. For more efficient and more robust feature matching, the authors plan to reduce the time cost by applying a parallel framework in the future.

**Author Contributions:** Conceptualization, X.H.; methodology, X.H.; software, X.H.; validation, X.H.; formal analysis, X.H.; investigation, X.H. and X.W.; resources, X.H.; data curation, X.H.; writing—original draft preparation, X.H. and X.W.; writing—review and editing, X.H. and D.P.; visualization, X.H. and D.P.; supervision, X.H.; project administration, X.H.; funding acquisition, X.H. All authors have read and agreed to the published version of the manuscript.

**Funding:** This research was funded by the National Natural Science Foundation of China (grant number 41701540), Basic Startup Funding of Sun Yat-sen University (grant number 76230-18841205), National Natural Science Foundation of China (grant number 41801386), Innovation Program of CSU, CAS (grant number Y8031831).

**Acknowledgments:** The authors would like to thank the National Laboratory of Pattern Recognition, Institute of Automation, Chinese Academy of Science for providing the close-range dataset. The authors would like to thank Yilong Han for calibrating camera distortions in the testing datasets.

**Conflicts of Interest:** The authors declare no conflict of interest.

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
