# Peer review of "Robust Feature Matching with Spatial Smoothness Constraints"

_remotesensing, doi:10.3390/rs12193158_

Round 1
Reviewer 1 Report
Feature matching is a fundamental issue in computer vision and photogrammetry applications. The local feature-based methods, e.g. SIFT, will generate false matches on images with viewpoint change or weak textrures. A graph-based matching method has been proposed in this manuscript to process the results of the local feature-based methods to improve the recall and precision performance. It was demonstrated in the experimental resutls that the proposed method works well. However, there are some issues that can be improved.
1. I suggest adding some references in the introduction, such as the following references [1]-[3], among which the references [1] is for images with repeatitive textures, and references [2] and [3] are to process the matching results of local feature-based methods.
[1] Chen M., Qin R., He H., et al., A Local Distinctive Features Matching Method for Remote Sensing Images with Repetitive Patterns, Photogrammetric Engineering and Remote Sensing, 2018, 84(8): 513-524.
[2] Ma J., Zhao J., Tian J., et al., Robust point matching via vector fifield consensus, IEEE Transactions on Image Processing, 2014, 23: 1706–1721.
[3] Li J., Hu Q., Ai M.. 4FP-Structure: A Robust Local Region Feature Descriptor, Photogrammetric Engineering and Remote Sensing, 2017, 83(12): 813-826.
2. It is recommended to improve the quality of some figures in the paper, such as Figure 1. The images are not very clear, and the description in the caption is also not clear enough.
3. I think it is better to add some comparative methods in the experiments. For example, the method proposed in [2] can be used for comparative analysis. The method in [2] is also for further processing of the matching results of methods such as SIFT, and the source code is provided by the authors. It can be easily transplanted into the experiment of this paper to conduct a fair comparative analysis with the same initial matching result as input.
Reviewer 2 Report
The manuscripts introduced a feature matching approach based on the similarity of surrounding feature points. The authors described their method clearly with illustrative figures. The experiments are sufficient. Their results demonstrate the performance of the proposed approach and support the conclusion. Therefore, I recommend to accept it for publication.
Reviewer 3 Report
Please give an explanation to selecting the the maximum and minimum cost (lines 133 and 139) for the implementation of your algorithm.
There exist some typos:
Sentense in lines 63-65 needs rephrasing according tomy reading.
change vison to vision in line 44
